

# Cirrus cloud optical and microphysical property retrievals from eMAS during SEAC4RS using bi-spectral reflectance measurements within the 1.88 μm water vapor absorption band

**K. Meyer[1,2], S. Platnick[2], G. T. Arnold[3,2], R. E. Holz[4], P. Veglio[4], J. Yorks[2], and C. Wang[5]**

[1]Goddard Earth Sciences Technology and Research (GESTAR), Universities Space Research Association, Columbia, Maryland, USA
[2]NASA Goddard Space Flight Center, Greenbelt, Maryland, USA
[3]Science Systems and Applications, Inc., Lanham, Maryland, USA
[4]Cooperative Institute for Meteorological Satellite Studies, University of Wisconsin – Madison, Madison, Wisconsin, USA
[5]Earth System Science Interdisciplinary Center, University of Maryland – College Park, College Park, Maryland, USA

**AMTD**

doi:10.5194/amt-2015-326

**Cirrus cloud optical and microphysical property retrievals**

K. Meyer et al.

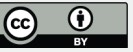

Received: 27 October 2015 – Accepted: 7 December 2015 – Published: 14 January 2016

Correspondence to: K. Meyer (kerry.meyer@nasa.gov)

Published by Copernicus Publications on behalf of the European Geosciences Union.

**AMTD**

doi:10.5194/amt-2015-326

**Cirrus cloud optical and microphysical property retrievals**

K. Meyer et al.

Discussion Paper | Discussion Paper | Discussion Paper | Discussion Paper | Discussion Paper |

**AMTD**

doi:10.5194/amt-2015-326

**Cirrus cloud optical and microphysical property retrievals**

K. Meyer et al.

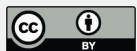

## Abstract

Previous bi-spectral imager retrievals of cloud optical thickness (COT) and effective particle radius (CER) based on the Nakajima and King (1990) approach, such as those of the operational MODIS cloud optical property retrieval product (MOD06), have typically paired a non-absorbing visible or near-infrared wavelength, sensitive to COT, with an absorbing shortwave or midwave infrared wavelength sensitive to CER. However, in practice it is only necessary to select two spectral channels that exhibit a strong contrast in cloud particle absorption. Here it is shown, using eMAS observations obtained during NASA's SEAC⁴RS field campaign, that selecting two absorbing wavelength channels within the broader 1.88 µm water vapor absorption band, namely the 1.83 and 1.93 µm channels that have sufficient differences in ice crystal single scattering albedo, can yield COT and CER retrievals for thin to moderately thick single-layer cirrus that are reasonably consistent with other solar and IR imager-based and lidar-based retrievals. A distinct advantage of this channel selection for cirrus cloud retrievals is that the surface contribution to measured cloudy TOA reflectance is minimized due to below-cloud water vapor absorption, thus reducing retrieval uncertainty resulting from errors in the surface reflectance assumption, as well as reducing the frequency of retrieval failures for thin cirrus clouds.

## 1  Introduction

Reflectance measurements within the water vapor absorption bands at 1.38 and 1.88 µm have been shown to be well suited for detecting cirrus clouds (Gao et al., 1993). This is because cirrus are typically located at high altitudes above the bulk of atmospheric water vapor, thus the contribution of the Earth's surface and boundary layer clouds to top-of-atmosphere (TOA) reflectance at wavelength channels within these bands is negligible in sufficiently moist atmospheric conditions due to absorption by the atmospheric water vapor below the cirrus layer. Moreover, TOA reflectance

of cirrus at 1.38 µm is sensitive to cloud optical thickness (COT), with only a small sensitivity to cloud effective particle radius (CER) due to weak ice crystal absorption (Kou et al., 1993; Yang et al., 2000). This sensitivity has been exploited using 1.38 µm reflectance measurements from the Moderate-resolution Imaging Spectroradiometer (MODIS) for retrieving the COT of thin cirrus clouds (Meyer et al., 2004, 2007; Meyer and Platnick, 2010). Thin cirrus are often problematic for traditional passive imager cloud retrievals, such as the operational MODIS cloud optical and microphysical property products (MOD06) (Platnick et al., 2003, 2015). Because the non-absorbing visible (VIS), near-infrared (NIR), or shortwave infrared (SWIR) wavelength channels typically used for COT retrievals, as well as the absorbing SWIR and mid-wave infrared (MWIR) wavelength channels used for CER retrievals, are sensitive to reflection by the underlying surface, such approaches are subject to larger retrieval uncertainty and increased frequency of retrieval failures for thin cirrus cases.

Previous 1.38 µm-based approaches, however, either require an a priori assumption about CER or necessitate the pairing of a second SWIR channel for simultaneous CER retrievals that can reintroduce surface sensitivity. Both cases can result in increased COT retrieval uncertainties, though it should be noted that surface sensitivity can be mitigated by pairing 1.38 µm with a channel centered at 1.88 µm (Gao et al., 2004). Here, a new approach is presented that pairs reflectance measurements at two narrow channels within the 1.88 µm water vapor absorption band to simultaneously retrieve cirrus COT and CER while minimizing the surface reflectance contribution. The retrieval has been applied to reflectance measurements from the Enhanced MODIS Airborne Simulator (eMAS) (King et al., 1996; Ellis et al., 2011). Retrieval results are shown for select case studies, as are comparisons with an eMAS-based version of MOD06, and retrievals from IR approaches and from the Cloud Physics Lidar (CPL) (McGill et al., 2002).

**AMTD**

doi:10.5194/amt-2015-326

**Cirrus cloud optical and microphysical property retrievals**

K. Meyer et al.



## 2   Data

The Enhanced MODIS Airborne Simulator (eMAS) (King et al., 1996), a line-scanning spectrometer deployed on NASA's high-altitude ER-2 research aircraft, measures radiances at 38 spectral channels in the wavelength range from 0.47 to 14.1 µm. With a maximum scan angle extending 43° to either side of nadir, eMAS observes 716 pixels across a 37 km wide ground swath at a nominal ER-2 altitude of 20 km, yielding pixel sizes on the Earth's surface of roughly 50 m at nadir. The ER-2 flew extensive science flights as part of the Studies of Emissions and Atmospheric Composition, Clouds and Climate Coupling by Regional Surveys (SEAC⁴RS) field campaign based in Houston, Texas, in August and September 2013, with a payload that included both the eMAS and Cloud Physics Lidar (CPL) within the same wing superpod. Numerous cirrus cloud scenes were observed during SEAC⁴RS, from which the present case studies are selected.

As part of normal field campaign efforts, the eMAS team provides cloud masking and cloud property retrieval products based on the operational MODIS cloud mask (MOD35) (Ackerman et al., 1998, 2008; Frey et al., 2008) and cloud top and optical property (MOD06) retrievals (Platnick et al., 2003). For SEAC⁴RS, these eMAS cloud products, referred to hereafter as MAS06, use the latest Collection 6 version of MOD06 that includes numerous algorithm updates and enhancements (Platnick et al., 2015), and also includes a cloud top retrieval from the NOAA Algorithm Working Group (AWG) PATMOS-x algorithm (based on the CLAVR-x algorithm used in Heidinger and Pavolonis, 2009). Note for SEAC⁴RS, the AWG PATMOS-x algorithm provides the default cloud top retrievals, and the cloud thermodynamic phase used by the cloud optical property retrievals is provided by the heritage MOD06 Collection 5 algorithm (King et al., 2006).

In addition to MAS06, two research-level infrared (IR) optimal estimation (OE) approaches have also been applied to eMAS for cirrus cloud retrievals. The first, referred to as FEANOR (Flexible Experimental Atmospheric Non-linear Optimal estimation Re-

Discussion Paper | Discussion Paper | Discussion Paper | Discussion Paper

**AMTD**

doi:10.5194/amt-2015-326

**Cirrus cloud optical and microphysical property retrievals**

K. Meyer et al.

trieval), uses the 8.5, 11, and 12 μm wavelength channels coupled with cloud top altitude prescribed from CPL, and provides retrievals of COT and CER (Veglio and Holz, 2015). Note for this investigation, FEANOR relies on mean IR radiances averaged over all co-located eMAS pixels within each CPL level 2 field-of-view, and the retrieval is only applied when CPL cloud top height (CTH) is above 8 km. The second approach, referred to here as OE-IR, also uses the 8.5, 11, and 12 μm channels, along with the 6.7, 7.2, 8.2, 12.6, 13.3, 13.6, and 13.9 μm channels, and provides full-swath pixel-level retrievals of COT, CER, and CTH (Wang et al., 2015a, b) at the native eMAS spatial resolution. Both FEANOR and OE-IR provide estimates of retrieval uncertainty that account for a variety of radiometric, ancillary, and model error sources.

The availability during SEAC[4]RS of CPL also allows for additional evaluation of the eMAS retrievals. CPL is an elastic backscatter lidar that was first deployed in 2000 (McGill et al., 2002) and has participated in over two-dozen field campaigns aboard the NASA ER-2 and Global Hawk aircrafts. CPL measures backscatter at three wavelengths, namely 355, 532, and 1064 nm, as well as depolarization at the 1064 nm wavelength. These lidar measurements enable a comprehensive analysis of the radiative and optical properties of cirrus clouds through parameters such as CTH, depolarization ratio, backscatter and extinction coefficients, and COT (McGill et al., 2004; Davis et al., 2010; Yorks et al., 2011). For the present investigation, the CPL curtain is co-located with near-nadir eMAS observations such that the respective COT and CTH retrievals can be compared.

## eMAS Calibration

For remote sensing science applications, absolute radiometric calibration is a critical component. Calibration of the eMAS thermal IR channels is monitored in-flight by viewing two onboard blackbody sources once every scan; the shortwave channels are calibrated in a laboratory setting pre- and post-deployment by observing AAF laboratory standard integrating hemispheres, with day-to-day fluctuations in the field monitored by a smaller portable hemisphere prior to each flight. In addition, because ambient

**AMTD**

doi:10.5194/amt-2015-326

**Cirrus cloud optical and microphysical property retrievals**

K. Meyer et al.

**AMTD**

doi:10.5194/amt-2015-326

**Cirrus cloud optical and microphysical property retrievals**

K. Meyer et al.

flight conditions are significantly different from those at ground level, yielding potential inconsistencies between the laboratory calibration and that at flight altitude, periodic underflights of Terra and Aqua MODIS are used as flight-level calibration sources via statistical comparisons of collocated reflectance measurements and cloud property re-trievals (e.g., King et al., 2010). Calibration is further characterized by post-campaign flights over vicarious calibration sites; for SEAC[4]RS, the site at Ivanpah Playa in Primm Valley, California, was used. The eMAS data used here include the latest available cal-ibration corrections derived from rigorous analysis of the available satellite underflights and vicarious calibration, and represent the eMAS team's best efforts at providing a SEAC[4]RS dataset suitable for scientific investigations (Arnold et al., 2014).

## 3 Methodology

Though eMAS does not include the 1.38 μm channel, three narrow channels located within the 1.88 μm water vapor absorption band are available. Figure 1 shows the spec-tral response functions of these channels, labeled B14, B15, and B16, and centered approximately at 1.83, 1.88, and 1.93 μm, respectively, plotted over the surface to TOA two-way transmittance calculated for a tropical ocean atmosphere using the Line-by-Line Radiative Transfer Model (LBLRTM) (Clough and Iacono, 1995), for a nadir view and overhead sun. It is evident that all three channels are located almost wholly within the broader absorption region, though the tails of the 1.83 and 1.93 μm channel re-sponse functions extend beyond the region of total attenuation. Surface effects are thus not completely screened at 1.83 and 1.93 μm as they are at 1.38 μm and the cen-tral 1.88 μm channel, even in moist atmospheres, though the contribution of surface reflection to TOA reflectance and retrieval uncertainty is substantially smaller than in the VIS/NIR/SWIR channels commonly used for COT and CER retrievals; moreover, contamination by low-altitude clouds is more likely than at 1.38 or 1.88 μm. However, thresholds on the central 1.88 μm channel reflectance (must be larger than 0.02) and the 1.88/0.65 μm channel reflectance ratio (must be larger than 0.09) are used here

Discussion Paper | Discussion Paper | Discussion Paper | Discussion Paper |

to identify and remove clear sky and low-altitude cloud pixels, respectively, that may otherwise be spuriously identified as thin cirrus using only the 1.83 and 1.93 μm channels. In addition, the case studies selected here only include ocean scenes for which the surface is dark, thus the contribution of surface reflection to measured TOA cirrus reflectance is expected to be negligible.

The 1.88 μm spectral region also exhibits markedly stronger ice crystal absorption than at 1.38 μm, and TOA reflectance is consequently more sensitive to particle size. The previous techniques utilizing 1.38 μm for single-channel cirrus COT retrievals (e.g., Meyer and Platnick, 2010), which require a priori assumptions of CER, will have much larger uncertainties when applied to the channels near 1.88 μm and are thus ill suited for this spectral region. However, a strong contrast in single scattering albedo ($\varpi_0$) is evident between the 1.83 and 1.93 μm channels, indicating stronger ice crystal absorption at the latter wavelength. Figure 2 shows $\varpi_0$ as a function of CER for the 1.83 μm (blue line) and 1.93 μm (green line) channels, as well as for the 1.38, 1.64, 2.1, and 3.79 μm MODIS channels (red, gold, light blue, and magenta dashed lines, respectively). The single scattering properties used here are for the severely roughened aggregate of hexagonal columns ice crystal habit (Yang et al., 2013) that was used to create the MOD06 and MAS06 ice cloud retrieval look-up tables (LUTs); note this ice crystal radiative model has been shown to provide better closure between VIS/NIR, IR, and lidar retrievals of cirrus COT (Holz et al., 2015). The contrast of $\varpi_0$ between 1.83 and 1.93 μm suggests the possibility of a bi-spectral retrieval technique for simultaneously inferring COT and CER for two absorbing channels in the manner of Nakajima and King (1990) and Platnick et al. (2001).

Figure 3 shows the bi-spectral dependence of 1.83 and 1.93 μm top-of-cloud reflectance on COT and CER when the cosines of the solar and view zenith angles are 0.9 and the relative azimuth angle is 120°. Here, spectral top-of-cloud reflectance is obtained from forward radiative transfer (RT) calculations, ignoring atmospheric gaseous absorption and assuming a black, non-reflecting surface, using the Discrete Ordinates Radiative Transfer (DISORT) algorithm (Stamnes et al., 1988). It is clear that 1.93 μm is

Discussion Paper | Discussion Paper | Discussion Paper | Discussion Paper |

**AMTD**

doi:10.5194/amt-2015-326

**Cirrus cloud optical and microphysical property retrievals**

K. Meyer et al.

quite sensitive to CER, and 1.83 µm is sensitive to thin to moderately thick COT, though it becomes insensitive roughly around COT = 20. In addition, because the 1.83 µm channel is also sensitive to CER, as shown by the plot of $\omega_0$ in Fig. 2, the LUT is largely non-orthogonal. While non-orthogonal LUTs are not ideal and imply larger re-

5 trieval uncertainties, the sensitivities of the two wavelengths are such that a bi-spectral retrieval can nevertheless be performed for cirrus clouds, which are often tenuous and less optically thick.

Like the 1.38 µm channel, however, the water vapor absorption that attenuates the surface reflection at 1.83 and 1.93 µm, thus allowing sensitivity to very thin cirrus

clouds, can also introduce biases in the measured cloudy sky TOA reflectances. This is because a non-negligible portion of atmospheric water vapor resides above cirrus clouds, which attenuates the measured cloudy TOA reflectances. Here, CTH from the AWG PATMOS-x algorithm now integrated into MAS06 is coupled with ancillary atmospheric profiles obtained from the National Centers for Environmental Prediction

(NCEP) Global Data Assimilation System (GDAS) 6-h "final run" archive analyses (Derber et al., 1991) to estimate the pixel-level above-cloud water vapor absorption at both 1.83 and 1.93 µm using the correlated $k$-distribution technique (e.g., Kratz, 1995). The respective atmospherically-corrected reflectances are then calculated and are used to infer COT and CER from pre-computed ice cloud LUTs derived under assump-

tions identical to MOD06/MAS06, i.e., using DISORT and the scattering properties of severely roughened aggregates of hexagonal columns (Yang et al., 2013), which are integrated over a modified Gamma size distribution (effective variance 0.1) as well as the appropriate eMAS spectral response functions.

**Retrieval Uncertainty**

The pixel-level retrieval solutions are found using Newtonian iteration to locate the minimum of a cost function defined in terms of the difference between the observed and forward-modeled LUT spectral reflectances; note no a priori is assumed, thus the cost function simplifies to the weighted least squares estimate (Rodgers, 1976; Heidinger

**AMTD**

doi:10.5194/amt-2015-326

Cirrus cloud optical and microphysical property retrievals

K. Meyer et al.

and Stephens, 2000; Cooper et al., 2003). A critical component of this approach is defining an appropriate estimate of measurement errors. The resulting measurement error covariance matrix is coupled with the forward-modeled Jacobian, or retrieval solution space sensitivity matrix (derived here from the forward-modeled retrieval LUTs), to provide a baseline retrieval uncertainty estimate that accounts for known error sources. Here, multiple error components are assumed to contribute to the total retrieval uncertainty, namely radiometric errors, atmospheric water vapor profile errors, and cloud model errors (specifically size distribution effective variance). Because the 1.83 and 1.93 μm channels are assumed to be nominally free of surface contamination in the over-ocean case studies shown here, uncertainty due to surface albedo error is not considered.

For eMAS, because ambient conditions at flight level are often not stable (a problem exacerbated by in-flight altitude changes), and can be substantially different from the laboratory conditions under which pre- and post-deployment calibration is typically performed, the absolute pixel-level radiometric uncertainty is unknown. Therefore a constant relative reflectance error, here 10 %, is assumed at both 1.83 and 1.93 μm; note for MAS06, reflectance errors are assumed to be 5 % for 3.7 μm, 10 % for 1.6 μm, and 7 % for the remaining channels. Water vapor profile errors are assumed to be 20 % at all atmospheric layers. For cloud model uncertainty, expected reflectance errors are estimated using forward RT calculations to determine TOA reflectance deviations due to changes in the effective variance (from 0.1 to 0.05 and 0.2) of the assumed ice particle size distributions used to integrate the single scattering properties of Yang et al. (2013). Note uncertainty due to an incorrect ice crystal habit assumption, which can vary widely in nature (van Diedenhoven et al., 2014) and is expected to contribute significantly to retrieval uncertainty yet in practice is difficult to quantify, is presently ignored, as it is in both MOD06 and MAS06.

**AMTD**

doi:10.5194/amt-2015-326

**Cirrus cloud optical and microphysical property retrievals**

K. Meyer et al.

## 4 Results

On the 18 September 2013 SEAC[4]RS science flight, the ER-2 overflew thin to moder-
ately thick cirrus over the Gulf of Mexico (flight track 8), as shown in the true color RGB
(0.65–0.55–0.47 µm) in Fig. 4a; the direction of travel of the ER-2 in this figure is from
top to bottom. AWG PATMOS-x CTH retrievals for this scene are shown in Fig. 4b. The
corresponding retrieved COT from MAS06 is shown in Fig. 4c; note MAS06 retrievals
for both ice and liquid phase clouds are shown, and can be identified by the dual phase
color bar at top right (warm colors for liquid, cool colors for ice). COT from the 1.83 µm
channel is shown in Fig. 4d. Disregarding any errors in the MAS06 cloud thermody-
namic phase discrimination, the 1.83 µm COT retrievals appear consistent with those
from MAS06. Given the identical cloud radiative model assumptions and forward RT
code used in both retrievals, this result is encouraging and bestows confidence in the
above-cloud water vapor attenuation correction.

Note also the larger spatial extent of the 1.83 µm COT retrievals compared to those of
MAS06. As implied by the CTH retrievals in Fig. 4b, the cloud mask evidently identifies
clouds throughout this scene, while the MAS06 COT retrievals imply large cloud-free
regions (gray color). Disregarding potential cloud mask errors, specifically false pos-
itive cloudy pixels, the cloud-free regions indicate MAS06 COT retrieval failures, i.e.,
the reflectance observations are outside of the LUT retrieval solution space. The larger
spatial extent of the 1.83 µm COT retrievals, however, indicates a lower occurrence of
retrieval failure, an expected result of the relative insensitivity of the 1.83 and 1.93 µm
channels to surface reflection, particularly for the case of optically thin cirrus clouds.
In addition, the RGB image implies the presence of low-altitude liquid phase clouds
underlying the cirrus layer. These clouds are also evident in the MAS06 COT image
Fig. 4c as the liquid phase retrievals in the cirrus-free portions of the track, as well
as the relatively large (i.e., bright green) COT features within the optically thinner por-
tions of the cirrus. Note, however, that these COT features within the thin cirrus are

**AMTD**

doi:10.5194/amt-2015-326

**Cirrus cloud optical and microphysical property retrievals**

K. Meyer et al.

Discussion Paper | Discussion Paper | Discussion Paper | Discussion Paper | Discussion Paper

Title Page

Abstract    Introduction

Conclusions    References

Tables    Figures

◀    ▶▮



not evident in the 1.83 µm COT image, implying potential multilayer cloud detection capabilities of reflectance measurements within the 1.88 µm water vapor band.

Conversely, CER, shown in Fig. 4e–h, exhibits less agreement in terms of retrieval magnitude, though the spatial CER patterns appear consistent. Here, CER retrievals are shown for the standard MAS06 CER channels, namely (panel e) 1.6 µm, (panel f) 2.1 µm, and (panel g) 3.7 µm, as well as for the 1.93 µm channel (panel h). Again, both liquid and ice phase MAS06 retrievals are shown, and can be identified by the dual phase color bar at bottom right. Disagreement between CER retrievals is not unexpected, in part because photon penetration depth within clouds has been shown to be spectrally dependent in the SWIR and MWIR (Platnick, 2000), though it is interesting to note that the 1.93 µm CER appears to have better agreement with the 3.7 µm CER. Similar to the COT retrievals, the presence of underlying low-altitude liquid phase clouds is evident in the MAS06 CER retrievals by the relatively small (purple) features within the optically thin portions of the cirrus, while these features are not evident in the 1.93 µm retrievals.

A comparison of nadir-view COT, CER, and CTH retrievals for the 13 September 2013 flight track of Fig. 4 is shown in Fig. 5; the earliest-time retrievals (left) correspond to the top of each panel in Fig. 4. Plotted in panel a are eMAS-based COT retrievals from MAS06 (red), the 1.83/1.93 µm channel pair (blue), the FEANOR IR optimal estimation technique (magenta), and the multi-channel OE-IR technique (gold) (Wang et al., 2015), as well as collocated 532 nm COT retrievals from CPL (green) (McGill et al., 2002); CER retrievals from MAS06 2.1 µm, FEANOR, OE-IR, and 1.83/1.93 µm are plotted in panel b. To assess the CTH assumption used for above-cloud water vapor attenuation correction, the MAS06 (AWG PATMOS-x) CTH retrievals are plotted in panel c along with those from CPL and OE-IR. The vertical bars for the eMAS-based retrievals in each panel denote $\pm 1\sigma$ retrieval uncertainty. Note each MAS06, OE-IR, and 1.83/1.93 µm point in this plot represents the mean retrieval over all eMAS pixels within the collocated CPL product footprint, and the respective retrieval fractions within each CPL footprint must be larger than 0.25 for inclusion here; as stated above,

**AMTD**

doi:10.5194/amt-2015-326

**Cirrus cloud optical and microphysical property retrievals**

K. Meyer et al.

Discussion Paper | Discussion Paper | Discussion Paper | Discussion Paper | Discussion Paper |

**AMTD**

doi:10.5194/amt-2015-326

**Cirrus cloud optical and microphysical property retrievals**

K. Meyer et al.

the FEANOR retrievals use the mean spectral IR radiances averaged over all eMAS pixels within the collocated CPL footprint. There is overall good agreement between all COT retrievals, in particular those from eMAS, and the CER retrievals, while divergent in some regions, nonetheless exhibit some overlap when considering the retrieval
uncertainties.

Figure 6 shows the full-swath eMAS retrievals for a later ER-2 segment (flight track 10) of the same 18 September 2013 SEAC$^4$RS science flight; the direction of travel of the ER-2 in this figure is again from top to bottom. Similar to Fig. 4, thin to moderately thick cirrus overlies the Gulf of Mexico, with scattered low-altitude liquid phase clouds
evident in some portions of the RGB image, as well as the MAS06 COT and CER retrieval images. Figure 7 shows the nadir-view COT, CER, and CTH retrievals for this track. As in Figs. 4 and 5, the COT retrievals all exhibit general agreement in magnitude and spatial patterns, while the CER retrievals exhibit less agreement, though the MAS06 and 1.93 µm CER retrievals have similar spatial patterns in both the full-swath
and nadir-view plots. In addition, the larger spatial extent of the 1.83/1.93 µm retrievals is evident in both figures, again indicating less frequent retrieval failures using this channel pair.

## 5   Discussion

Previous bi-spectral imager retrievals of cloud optical thickness (COT) and effective
particle radius (CER) based on the Nakajima and King (1990) approach, such as those of the operational MODIS cloud optical property product (MOD06), have typically paired a non-absorbing VIS or NIR wavelength channel, sensitive to COT, with an absorbing SWIR or MWIR wavelength channel sensitive to CER. However, TOA reflectance measurements in these spectral channels can be quite sensitive to contributions from
surface reflection, in particular for the case of optically thin cirrus clouds. Thus cirrus retrieval approaches that rely on these channels are often subject to larger retrieval uncertainty and increased retrieval failure frequency (i.e., reflectance observations that

are outside the retrieval solution space) since they require appropriate assumptions regarding spectral surface reflection.

In practice it is only necessary to select two spectral channels that exhibit a strong contrast in cloud particle absorption. Here it is shown that two absorbing wavelength channels within the broader 1.88 µm water vapor absorption band, namely the 1.83 and 1.93 µm channels, have sufficient differences in ice crystal single scattering albedo such that a bi-spectral COT-CER retrieval approach can be applied. A distinct advantage of this channel selection for cirrus cloud retrievals is that the surface contribution to measured cloudy TOA reflectance in these channels is minimized due to below-cloud water vapor absorption, thus reducing retrieval uncertainty due to errors in the surface reflection assumption as well as reducing the occurrence of retrieval failures. Using two cirrus cloud case studies observed by eMAS over the Gulf of Mexico during NASA's SEAC[4]RS field campaign, it is shown that the 1.83/1.93 µm channel pair can yield COT and CER retrievals for thin to moderately thick single-layer cirrus that are reasonably consistent with other solar and IR imager-based retrievals, as well as lidar-based COT retrievals from collocated CPL. It is also shown that the present approach can provide useful information in multilayer cloud cases, i.e., cirrus overlying low-altitude liquid clouds, again due to the below-cirrus water vapor absorption that results in the reduced sensitivity of TOA reflectance at 1.83 and 1.93 µm to low-altitude clouds.

Finally, it is worth reemphasizing that, unlike the 1.38 and central 1.88 µm wavelength channels, below-cirrus atmospheric water vapor absorption does not completely attenuate the contribution of surface reflection in the 1.83 and 1.93 µm channels (see Fig. 1). Nevertheless, the surface contribution is substantially smaller than that in the VIS/NIR/SWIR window channels commonly used for COT and CER retrievals. For the case studies shown here, scenes over dark ocean were intentionally selected such that the contribution of surface reflection to the measured TOA reflectance at 1.83 and 1.93 µm is negligible. A more general application of the present technique over all surface types requires reasonable assumptions for surface reflection at 1.83 and 1.93 µm,

# AMTD

doi:10.5194/amt-2015-326

## Cirrus cloud optical and microphysical property retrievals

K. Meyer et al.

Discussion Paper | Discussion Paper | Discussion Paper | Discussion Paper

even though the surface contribution nonetheless is greatly minimized. Such efforts, however, are left for future investigations.

*Acknowledgements.* The authors would like to thank Gala Wind for her extensive development of the shared-core retrieval code used by the operational MOD06 products and its application to other space-borne and airborne sensors such as eMAS. The authors would also like to thank Nandana Amarasinghe for his efforts toward enhancing our forward radiative transfer modeling capabilities. This research was supported by the NASA Radiation Sciences Program for participation in the SEAC$^4$RS field campaign and by Atmospheric Composition Campaign Data Analysis and Modeling funding (NASA grant NNX15AD44G, PI Bastiaan van Diedenhoven). The eMAS level-1 data used in this investigation are publicly available, and were obtained from the NASA Level 1 and Atmosphere Archive and Distribution System (LAADS) (http://ladsweb.nascom.nasa.gov); the CPL data are courtesy of the CPL science team (Matt McGill and co-author John Yorks).

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

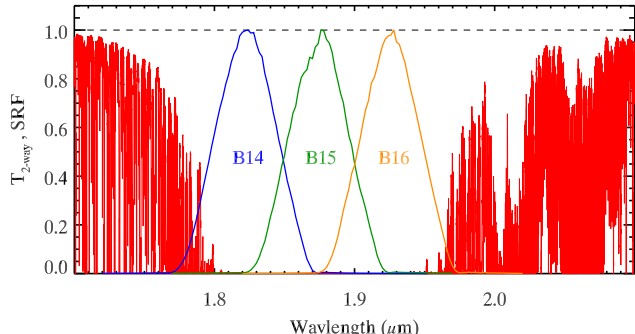

**Figure 1.** Spectral two-way transmittance, from TOA to surface, calculated with LBLRTM using a tropical ocean atmosphere. Spectral response functions during the SEAC4RS campaign for eMAS bands 14, 15, and 16 (band centers at approximately 1.83, 1.88, and 1.93 µm, respectively) are also shown.

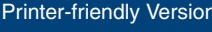



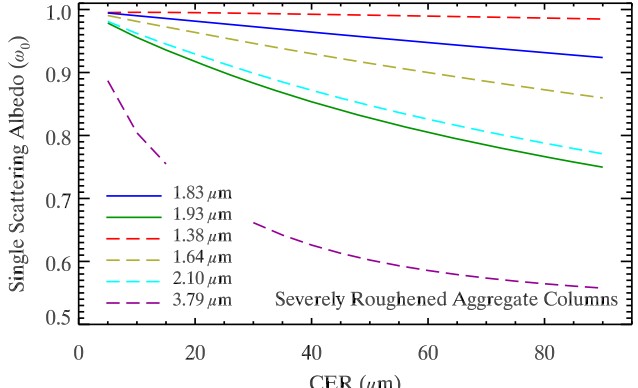

**Figure 2.** Bulk single scatter albedo ($\varpi_0$) for severely roughened aggregate hexagonal column ice crystals as a function of cloud effective particle radius (CER) for the eMAS 1.83 and 1.93 μm channels (blue and green, respectively) and the MODIS 1.38, 1.64, 2.1, and 3.79 μm channels (dotted red, gold, light blue, and magenta, respectively).

**AMTD**

doi:10.5194/amt-2015-326

**Cirrus cloud optical and microphysical property retrievals**

K. Meyer et al.

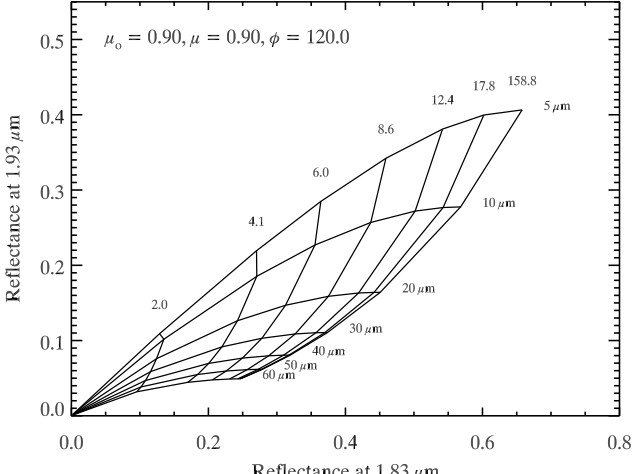

**Figure 3.** Two-channel plot illustrating the sensitivity of the 1.83 and 1.93 μm eMAS channels to cloud optical thickness (near-vertical lines) and effective particle radius (near-horizontal lines), respectively.

Discussion Paper | Discussion Paper | Discussion Paper | Discussion Paper | Discussion Paper |

**AMTD**

doi:10.5194/amt-2015-326

**Cirrus cloud optical and microphysical property retrievals**

K. Meyer et al.

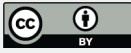

# AMTD

doi:10.5194/amt-2015-326

**Cirrus cloud optical and microphysical property retrievals**

K. Meyer et al.

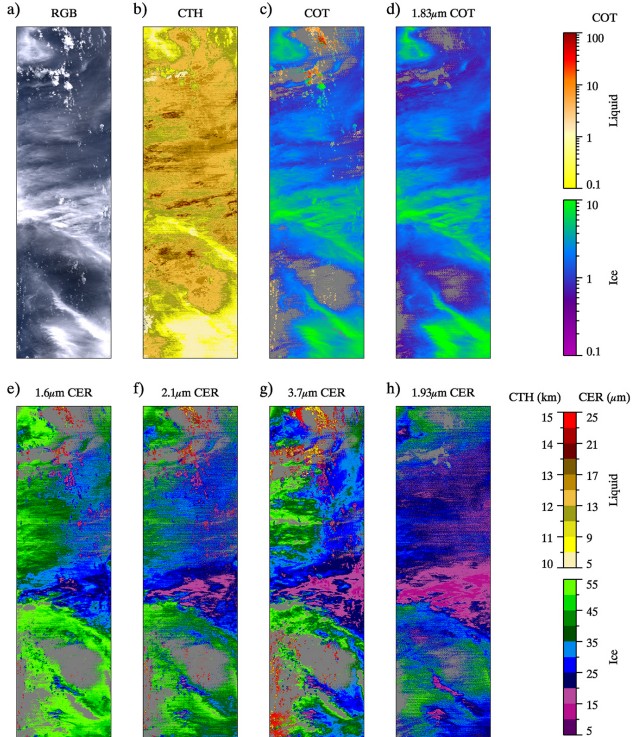

**Figure 4.** eMAS CTH, COT, and CER retrievals for a portion of track 8 of the 18 September 2013 SEAC$^4$RS science flight. The direction of travel of the ER-2 is from top to bottom in each panel. **(a)** True color RGB (0.65–0.55–0.47 µm).

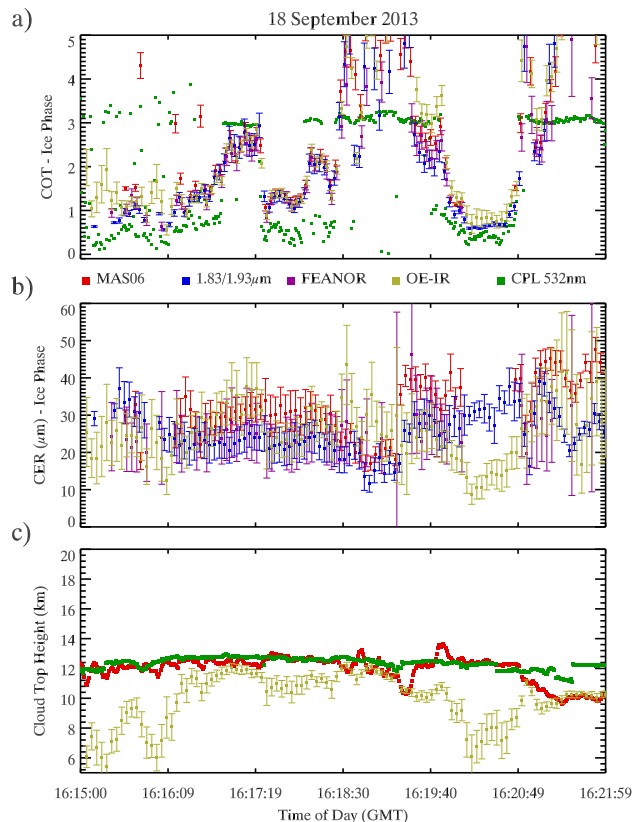

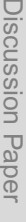

# AMTD

doi:10.5194/amt-2015-326

**Cirrus cloud optical and microphysical property retrievals**

K. Meyer et al.

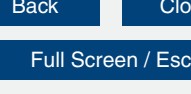

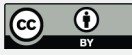

**Figure 5.** A comparison of nadir-view ice phase COT **(a)**, CER **(b)**, and CTH **(c)** retrievals for the same track as in Fig. 6; the earliest-time retrievals correspond to the top of each panel in Fig. 6. Vertical bars indicate estimated retrieval uncertainties. The retrievals plotted here are from MAS06 (red), the 1.83/1.93 µm approach (blue), FEANOR optimal estimation (magenta), and OE-IR (gold), as well as collocated CPL 532 nm (green); note the MAS06 CER retrievals are those using the 2.1 µm channel.

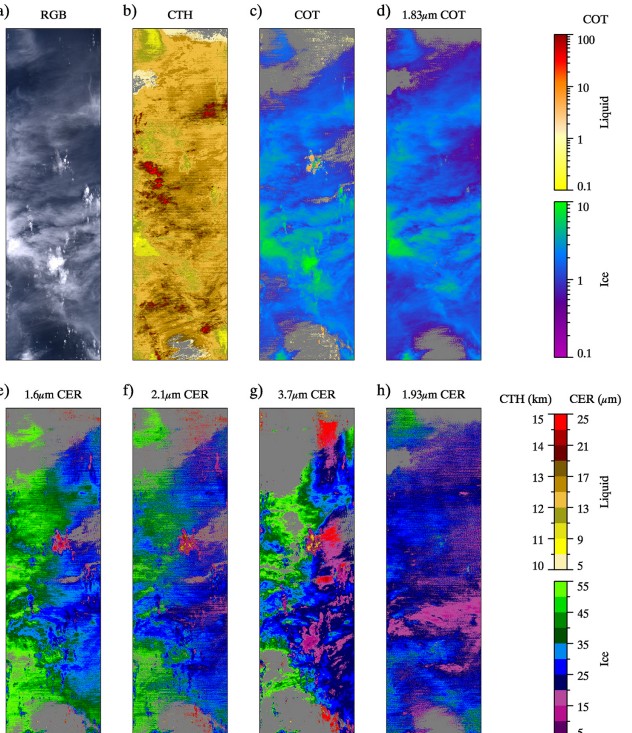

**Figure 6.** Same as Fig. 4, but for a portion of track 10 of the 18 September 2013 SEAC[4]RS science flight. The direction of travel of the ER-2 is again from top to bottom in each panel.

**AMTD**

doi:10.5194/amt-2015-326

**Cirrus cloud optical and microphysical property retrievals**

K. Meyer et al.

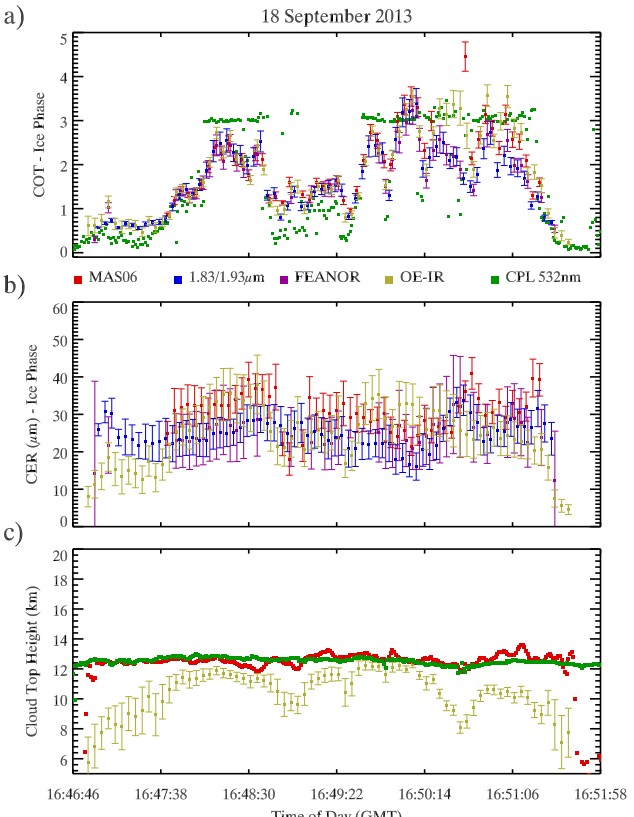

**Figure 7.** Same as Fig. 5, but for the track shown in Fig. 6.

**AMTD**

doi:10.5194/amt-2015-326

**Cirrus cloud optical and microphysical property retrievals**

K. Meyer et al.

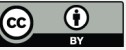