# Peer review of "Cirrus cloud optical and microphysical property retrievals from eMAS during SEAC4RS using bi-spectral reflectance measurements within the 1.88 μm water vapor absorption band"

_Atmospheric Measurement Techniques, 2015_

## Referee Comment (RC1) · Anonymous Referee #1 · 9 Feb 2016

General comments:

The well-known Nakajima-King bi-spectral algorithm for simultaneously retrieving cloud optical thickness and effective particle size is usually implemented by using a visible channel and a near-infrared channel, say, two channels centered at 0.66 $\mu$m and 2.1 $\mu$m. This study explores the applicability of two channels within a water-vapor-absorbing band. To be more specific, two channels at 1.83 $\mu$m and 1.93 $\mu$m are used. The observations made by the Enhanced MODIS Airborne Simulator (eMAS) deployed during the SEAC4RS field campaign are used in this study. The merit of using the 1.83-

and 1.93-$\mu$m channels is that the two channels are not sensitive to surface reflection because of the absorption of water vapor in the lower atmosphere below the cloud.

Overall, the manuscript is well organized and clearly written. No major technical errors were found. The manuscript is recommended for publication after some minor revisions are made.

Specific comments:

1. As explained in the manuscript, it is critical to account for the absorption of the water vapor above cirrus clouds when implementing the 1.83 $\mu$m/1.93 $\mu$m bi-spectral retrieval algorithm. Further, the correlated k-distribution (CKD) method was used in the forward model to correct the water vapor absorption. The manuscript cites the work by Kratz (1995) for the CKD simulation. But Kartz (1995) did not consider the spectral response function. A recent paper (Liu et al., 2015: A fast Visible Infrared Imaging Radiometer Suite simulator for cloudy atmospheres, J. Geophys. Res. Atmos., 120, doi:10.1002/2014JD022443) fully considers the responses function. It is suggested that the aforesaid paper be cited.

The authors are suggested to provide a paragraph to explain the incorporation of the spectral response function into the CKD simulation.

2. Can an empirical approach be used to correct water vapor absorption above ice clouds? For example, to infer cirrus reflectance using MODIS 1.375 channel, an empirical method is used to remove the effect of the water vapor above cirrus clouds. Can the aforesaid MODIS empirical approach, after some modifications, be applied to the eMAS 1.83- and 1.93-$\mu$m bands to remove the absorption of the above-cloud water vapor?

3. The cloud effective radius (CER) values corresponding to the 1.93 $\mu$m channel and the 1.6-$\mu$m or 2.1-$\mu$m channel are quite different. For downstream applications (e.g., the assessment of cloud radiative forcing), which CER value should be used

in order to obtain an optimal assessment? Note, in cloud radiation parameterization used in radiative transfer scheme involved in GCMs, the asymmetry factor and single-scattering albedo are parameterized in terms of CER. Thus, using optimal CER values in radiative transfer simulations is critical.

4. Page 7, line 6 from bottom: "all three channels are located almost wholly" to "B15 is located almost wholly".

5. Page 8, line 2: "is more likely than" should be "is more likely larger than"

6. Page 14, line 7 from bottom: "Fig. 4 and 5" should be "Figs. 4 and 5".

---

## Referee Comment (RC2) · Anonymous Referee #2 · 11 Feb 2016

General comments:

The work presented in this paper focuses on an alternative dual-channel technique to retrieve thin cloud properties from airborne spectral measurments. The paper is well written and can be of interest for any reader willing to implement a similar method with other instruments. The technical procedure is described in details and leaves no room for scientific ambiguity.

Final publication in AMT should be obviously warranted, since just few minor changes would be needed to improve clarity also for the non-experienced reader. However, one

major issue is still unclear and namely: why have land pixels been discarded in the analysis?

In Page 8 line 163 and following it is stated that: *In addition, the case studies selected here only include ocean scenes for which the surface is dark, thus the contribution of surface reflection to measured TOA cirrus reflectance is expected to be negligible*

But in the abstract the algorithm is termed to be very effective in surface screening and in delivering more accurate results. I would have expected the inclusion of a bright scene case study along with the analysis of a dark water surface. How can be the effectiveness of the algorithm be judged if the underlying surface is barely reflective?

Even if it can be clearly guessed by the experienced reader that the influence of the surface albedo is minimized, I think that it should be made explicit. Therefore I suggest to investigate the dependency of the 1.83/1.93 mkm retrieval technique also as function of surface reflectivity with an another case study and provide a table with relative errors in COT and CER among the presented algorithms and methods.

Minor comments:

p7 l155: Only the range 1.83-1.93 mkm is shown in Figure 1. It would be nice to show also the 1.38 mkm for illustrative purposes.

p12 l254: Notwithstanding the remarkable resemblance between image c) and d), could you please provide a justification of neglecting the error in the cloud phase discrimination? I think it is important for the reader to understand the assumption behind this choice.

Why aren't low-level warm clouds - in a) the brightest cumulus and in c) the red-coded at the top scans - showing up in d) too? Is it because of the discussion on the coverage of the LUT (p12 l264) or because of a suboptimal flagging?

p13 l294 and ff: The discussion on plot c) on CTH. It seems to me that the OE-IR mehtod delivers considerably correlated results across variables, namely CTH, CER

and COT. Could you provide more details on this, even if OE-IR is not really the focus of the present manuscript?

Additional details on the cloud phase (discrimination, see above) also would be appreciated, because when looking at Fig.5-c, is difficult to see whether the green and red lines are all in ice and not mixed-phase.

---

## Author Comment (AC1) · 4 Apr 2016

The authors would like to thank the reviewer for the careful evaluation of the manuscript and constructive comments. Our responses to the specific comments are below.

1. As explained in the manuscript, it is critical to account for the absorption of the water vapor above cirrus clouds when implementing the 1.83 μm/1.93 μm bi-spectral retrieval algorithm. Further, the correlated k-distribution (CKD) method was used in the forward model to correct the water vapor absorption. The manuscript cites the work by Kratz (1995) for the CKD simulation. But Kartz (1995) did not consider the spectral response function. A recent paper (Liu et al., 2015: A fast Visible Infrared Imaging Radiometer Suite simulator for cloudy atmospheres, J. Geophys. Res. Atmos., 120, doi:10.1002/2014JD022443) fully considers the responses function. It is suggested that the aforesaid paper be cited.

   The authors are suggested to provide a paragraph to explain the incorporation of the spectral response function into the CKD simulation.

   > **Response:** This is an excellent comment. Kratz indeed did not consider the spectral response function (SRF) in his CKD approach, but instead selected uniformly weighted spectral widths that yielded band-averaged transmittance roughly equivalent to more accurate line-by-line calculations that do consider the SRF. Moreover, for the present investigation, we have adopted Kratz's MODIS-specific 1.38μm CKD routine for the above-cloud atmospheric absorption correction in both the 1.83 and 1.93μm eMAS channels. We concede that at first glance this appears to be an incompatible application of this specific CKD routine. However, comparisons with more exact line-by-line calculations using LBLRTM (Clough et al., 1992; Clough and Iacono, 1995; Clough et al., 2005), in which we do explicitly account for the eMAS SRFs when calculating band-averaged transmittance, show that the MODIS 1.38μm CKD compares well with the line-by-line 1.83 and 1.93μm band-averaged two-way transmittances from TOA down to roughly 10-11km altitude (note that the case studies shown in the paper have CPL-retrieved cloud top altitudes of roughly 12km). This is evident in the plot below, which shows two-way transmittance calculated from the 1.38μm CKD (black line), as well as that calculated from LBLRTM (accounting for the exact SRF) at 1.83μm (red line), 1.88μm (blue line), and 1.93μm (gold line). All calculations shown here assume a tropical ocean atmosphere and solar and sensor zenith angles of 60°.

   > For on-line atmospheric absorption/transmittance calculations within a retrieval algorithm that must account for pixel-level changes in atmospheric profiles, cloud altitudes, etc., the CKD approach is preferable because it is significantly more efficient than the computationally expensive line-by-line calculations. However, because we were not aware of existing CKD routines (or other efficient techniques) specific to the eMAS channels within the 1.88μm water vapor absorption band, and because the 1.38μm CKD appears to sufficiently account for the absorption above typical cirrus cloud altitudes, we therefore opted for the Kratz 1.38μm CKD, and feel that it is adequate for the present proof-of-concept investigation. We do note, however, that work is ongoing, specifically by the research group of colleague P. Yang at Texas A&M, to develop an efficient atmospheric transmittance approach

that is capable of accounting for instrument-specific SRFs; we hope to implement this technique within the present cirrus retrieval algorithm in the future, though it is unfortunately not available to us at this time.

Regarding the reviewer's suggested paragraph addition, we think it is unnecessary to go into that level of detail. In fact, we fear it may confuse the issue for the reader to add a discussion of the selection and use of the MODIS 1.38μm CKD and comparisons with LBLRTM. While we readily admit the fact that this CKD routine works for the present application is quite fortuitous, the ultimate goal of this paper is to introduce the retrieval concept and show that it is viable. Finally, we would like to alert the reviewer that we have, at the request of the editor, provided additional details in Section 3 regarding how the CKD is used here to estimate the above-cloud water vapor transmittance (see p. 10, lines 218-230).

[Figure]

2. Can an empirical approach be used to correct water vapor absorption above ice clouds? For example, to infer cirrus reflectance using MODIS 1.375 channel, an empirical method is used to remove the effect of the water vapor above cirrus clouds. Can the aforesaid MODIS empirical approach, after some modifications, be applied to the eMAS 1.83- and 1.93-μm bands to remove the absorption of the above-cloud water vapor?

> **Response:** Another excellent question. We actually did attempt to develop an empirical approach similar to the 1.38μm *Meyer and Platnick* (2010) approach that couples 1.24μm with 1.38μm for estimating above-cloud water vapor absorption. This was during our initial attempts to modify the *Meyer and Platnick* cirrus optical thickness retrieval for use with the eMAS 1.88μm channel. We found, however,

that eMAS does not have a solar window channel that can be coupled with 1.88μm that is analogous to the 1.24/1.38μm combination; note that ice crystal absorption is stronger at 1.88μm compared to that at 1.38μm (see the plot of single-scattering albedo in Fig. 2 in the text), thus 1.24μm, even if available on eMAS (it isn't), is not an appropriate partner for such an approach, nor are the visible channels used by *Gao et al.* (2002) that we believe the reviewer is referring to. In fact, it was during these attempts to implement the *Meyer and Platnick* technique that we discovered the utility of the 1.83/1.93μm channel pair for COT/CER retrievals.

3. The cloud effective radius (CER) values corresponding to the 1.93 μm channel and the 1.6-μm or 2.1-μm channel are quite different. For downstream applications (e.g., the assessment of cloud radiative forcing), which CER value should be used in order to obtain an optimal assessment? Note, in cloud radiation parameterization used in radiative transfer scheme involved in GCMs, the asymmetry factor and single-scattering albedo are parameterized in terms of CER. Thus, using optimal CER values in radiative transfer simulations is critical.

> **Response:** This is a very good question that we believe can be asked of all imager-based size retrievals using either solar or IR channels, as there is often little agreement amongst them (e.g., Figs. 5b and 7b in the text). Moreover, there is good reason to expect these retrievals to be different, for instance due to potentially differing vertical sensitivities within the cloud (see, e.g., *Platnick*, 2000, for examples of liquid water cloud vertical weighting functions). In fact, 1.93μm may be weighted more towards the top of the cloud due to stronger ice crystal absorption than at 1.6 and 2.1μm (see Fig. 2 in the text), coupled with in-cloud water vapor absorption that further attenuates the reflected signal. That said, we believe that providing a meaningful answer to this question is well beyond the scope of the present investigation, since appropriately addressing the sensitivities of the CER-sensitive solar and IR channels will require extensive forward modeling of clouds with realistic microphysical and 3D structures. While we abstain from addressing these issues here, we nevertheless appreciate the reviewer's thought-provoking question.
>
> We would also like to point out that subsequent to submitting this paper, an error was found in the MAS06 ice cloud retrieval LUTs specific to the 1.6μm CER retrievals. This error has since been addressed, and we have updated Figs. 4 and 6 accordingly; note that the LUT correction results in smaller 1.6μm CER retrievals that are now "in family" with the 2.1μm CER retrievals.

4. Page 7, line 6 from bottom: "all three channels are located almost wholly" to "B15 is located almost wholly".

> **Response:** Done.

5. Page 8, line 2: "is more likely than" should be "is more likely larger than"

> **Response:** Done.

6. Page 14, line 7 from bottom: "Fig. 4 and 5" should be "Figs. 4 and 5".

   **Response:** Done.

**References**

Clough, S. A., Iacono, M. J., and Moncet, J.-L.: Line-by-line calculation of atmospheric fluxes and cooling rates: Application to water vapor, J. Geophys. Res., 97, 15761-15785, 1992.

Clough, S. A., Shephard, M. W., Mlawer, E. J., Delamere, J. S., Iacono, M. J., Cady-Pereira, K., Boukabara, S., and Brown, P. D.: Atmospheric radiative transfer modeling: A summary of the AER codes, J. Quant. Spectrosc. Radiat. Transfer, 91, 233-244, 2005.

Gao, B.-C., Yang, P., Han, W., Li, R.-R., and Wiscombe, W.: An algorithm using visible and 1.38μm channels to retrieve cirrus cloud reflectances from aircraft and satellite data, IEEE Trans. Geosci. Remote Sens., 40, 1659-1668, 2002.

Platnick, S.: Vertical photon transport in cloud remote sensing problems, J. Geophys. Res., 105, 22919-22935, 2000.

---

## Author Comment (AC2) · 4 Apr 2016

The authors would like to thank the reviewer for the careful evaluation of the manuscript and constructive comments. Our responses to the general and specific comments are below.

**General Comments:**

Final publication in AMT should be obviously warranted, since just few minor changes would be needed to improve clarity also for the non-experienced reader. However, one major issue is still unclear and namely: why have land pixels been discarded in the analysis?
In Page 8 line 163 and following it is stated that: *In addition, the case studies selected here only include ocean scenes for which the surface is dark, thus the contribution of surface reflection to measured TOA cirrus reflectance is expected to be negligible*
But in the abstract the algorithm is termed to be very effective in surface screening and in delivering more accurate results. I would have expected the inclusion of a bright scene case study along with the analysis of a dark water surface. How can be the effectiveness of the algorithm be judged if the underlying surface is barely reflective?
Even if it can be clearly guessed by the experienced reader that the influence of the surface albedo is minimized, I think that it should be made explicit. Therefore I suggest to investigate the dependency of the 1.83/1.93 mkm retrieval technique also as function of surface reflectivity with an another case study and provide a table with relative errors in COT and CER among the presented algorithms and methods.

> **Response:** These are excellent comments. We would first like to state that we did not claim in the abstract that the surface is completely screened from the TOA reflectance in the $1.83\mu m$ and $1.93\mu m$ channels, only that its contribution is minimized. In fact, we explicitly state in Section 3 (p. 7-8, lines 165-173) and in the discussion (Section 5) that surface effects are not completely screened at $1.83\mu m$ and $1.93\mu m$. The minimization of the surface contribution in these channels is of course in comparison with the heritage solar window channels used in the current MODIS and eMAS cloud optical retrievals, in which the surface can be expected to be a large contributor to TOA reflectance for optically thinner clouds such as cirrus. For the heritage window channel algorithms, surface albedo errors can not only cause increased retrieval uncertainty for optically thin cirrus cases, but can often cause the retrievals to fail outright. To eliminate any confusion, we have modified the abstract to clarify this statement (p. 2, line 37-41); we have also modified the first sentence of the introduction (p. 3, line 52).
>
> Our reasoning for excluding land cases from the present analysis is in part due to the potentially non-negligible surface contribution to TOA reflectance at $1.83\mu m$ and $1.93\mu m$ in arid atmospheres or over some bright land surfaces. For instance, assuming a standard mid-latitude winter atmosphere over land having column water vapor amount on the order of $1.2g\ m^{-2}$, the TOA to surface two-way transmittance at nadir solar and viewing angles, calculated using LBLRTM (Clough et al., 1992; Clough and Iacono, 1995; Clough et al., 2005) and averaged over the eMAS spectral response functions, is roughly 8% at $1.83\mu m$ and 3% at $1.93\mu m$. Assuming a bright Lambertian surface albedo of 10%, this corresponds to a contribution to

TOA reflectance in clear sky conditions of roughly 0.008 at 1.83µm and 0.003 at 1.93µm; certainly small contributions, but potentially non-negligible for optically thin cirrus that can have TOA reflectances less than 0.03. Furthermore, while we do possess the capability to add an appropriate surface albedo to the cloud optical retrieval look-up tables, as is currently done for land surfaces in MOD06 and MAS06, we do not currently have the capability to efficiently model the below-cloud atmospheric water vapor absorption on-line within the retrieval algorithm, though we hope to add this capability in the future (see our response to the first comment of Reviewer 1). Thus these retrievals cannot at present be implemented over land unless the surface is assumed to be completely obscured. Finally, the desire to judge the present approach against the standard MAS06 algorithms, as well as the IR-based algorithms, in cases in which we believe these algorithms actually have some skill, i.e., over dark surfaces, further influenced our decision.

**Specific Comments:**

p7 l155: Only the range 1.83-1.93 mkm is shown in Figure 1. It would be nice to show also the 1.38 mkm for illustrative purposes.

**Response:** While we share the reviewer's interest in the 1.38µm channel, we have nevertheless chosen to leave Fig. 1 unchanged (with the exception of modifying the line colors) because extending it to include 1.38µm obscured the finer details of the absorption spectra. We are however including a plot of the entire spectral region here for the reviewer's benefit (below; blue dotted line denotes the MODIS 1.38µm spectral response function); the atmospheric profile and view/solar angles match those shown in Fig. 1. We also note that the effect of the details of the 1.38µm channel are in part shown by the single-scattering albedo plot in Fig. 2.

[Figure]

p12 l254: Notwithstanding the remarkable resemblance between image c) and d), could you please provide a justification of neglecting the error in the cloud phase discrimination? I think it is important for the reader to understand the assumption behind this choice.

> **Response:** We should first reemphasize that the cloud phase algorithm used in the MAS06 retrievals is essentially the MOD06 C5 phase (King et al., 2006) (see p. 5, lines 108-110) that was known to have issues, e.g., identifying thin cirrus as liquid phase, etc. (see, e.g., Marchant et al., 2016). Moreover, there is no cloud phase decision filtering applied to the 1.83/1.93µm retrievals such that cases where the MAS06 phase is incorrectly liquid, or multilayer cases (thin cirrus overlying thick liquid) that are identified as liquid (in some cases correctly in a radiative sense, at least at the window channels used by MAS06), are included in the retrieval pixel population. This decision was made in order to highlight the capabilities of the water vapor absorption channels for retrieving optically thin cirrus clouds that are problematic for the window channel-based retrievals (e.g., phase discrimination errors, failed retrievals) as well as for retrieving only the overlying ice cloud in multilayer ice-over-liquid cases (which is likely the case in portions of the scene at issue here). We do note that work is ongoing to implement the MOD06 C6 phase, which has been shown to provide better phase discrimination skill compared with C5 for MODIS via comparisons with the CALIPSO lidar CALIOP (Marchant et al., 2016), within the MAS06 airborne retrievals.
>
> That said, we acknowledge that phase discrimination errors are not part of the error budgets for either the MAS06 retrievals or the present 1.83/1.93µm retrievals. The idea we are conveying with this statement is that for the pixels that MAS06 does identify as ice phase and provides successful COT/CER retrieval pairs, the COT retrievals from the two approaches are consistent with each other. It is worth reemphasizing that the primary goal of the current investigation is to be a proof of concept, and we believe this is clearly demonstrated by the results shown.

Why aren't low-level warm clouds - in a) the brightest cumulus and in c) the red-coded at the top scans - showing up in d) too? Is it because of the discussion on the coverage of the LUT (p12 l264) or because of a suboptimal flagging?

> **Response:** This is a good observation, and we in fact identified these clouds in the text discussion [page 14, lines 313-318]. As we stated above, we do not apply any filtering based on the MAS06-retrieved cloud phase. Instead, in addition to using the MAS06 cloud mask to identify cloudy pixels, we apply thresholds to the reflectance in the central 1.88µm channel (must be larger than 0.02) and to the 1.88/0.66µm reflectance ratio (must be larger than 0.09) (see p. 8, lines 174-178). These thresholds act to exclude clear sky pixels that are misidentified as cloudy sky, a situation that occurs primarily over bright land surfaces, and those cloudy pixels having only lower-altitude liquid phase clouds with insufficient above-cloud water vapor such that they can contribute to the TOA reflectance, in which case the clouds will be visible at 1.88µm though much brighter at 0.66µm.

p13 l294 and ff: The discussion on plot c) on CTH. It seems to me that the OE-IR mehtod delivers considerably correlated results across variables, namely CTH, CER and COT. Could you provide more details on this, even if OE-IR is not really the focus of the present manuscript?

> **Response:** The reviewer makes a good observation that the OE-IR retrievals appear to be correlated across all variables, at least for the case studies shown here. While this is certainly an interesting result, the essential question is whether this correlation represents real physics and/or algorithm issues, the answer to which is difficult to sort out and is quite beyond the scope of the present paper. We emphasize that these these retrievals are included only as an additional point of reference for the 1.83/1.93μm approach, and thus believe that providing specific algorithm details is not necessary for the present paper, though we do note that the wavelength channels used by OE-IR are provided in Section 2 (see p. 5-6, lines 119-120). We would like to point the reviewer to the Wang et al. manuscripts for further details of the OE-IR retrievals once they are published (at the time of this writing Part II is in press, Part I remains under review).

Additional details on the cloud phase (discrimination, see above) also would be appreciated, because when looking at Fig.5-c, is difficult to see whether the green and red lines are all in ice and not mixed-phase.

> **Response:** This is a good comment. When creating the curtain plots in Figs. 5 and 7, we do impose a cloud phase filter on the MAS06 (using the same MAS06 optical property retrieval phase used to discriminate ice and liquid clouds in Figs. 4 and 6) and OE-IR (using the IR-derived cloud phase of Baum et al. (2010)) retrievals, such that only ice phase pixels are used; the 1.83/1.93μm retrievals do not impose a cloud phase filter. The FEANOR retrieval, which is run on mean IR radiances averaged over all eMAS pixels within the CPL field-of-view, is implicitly ice phase only since it is only applied when CPL cloud top height is above 8km. We have added text clarifying these details for OE-IR (p. 6, lines 122-123) and for Fig. 5 (p. 15, lines 348-353). The criteria for FEANOR retrieval application was previously included in Section 2 (see p. 5, lines 116-118), and the pixel selection criteria for the 1.83/1.93μm retrievals was previously included in Section 3 (p. 8, lines 174-178).

**References**

Baum, B. A., Menzel, W. P., Frey, R. A., Tobin, D. C., Holz, R. E., Ackerman, S. A., Heidinger, A. K., and Yang, P.: MODIS cloud-top property refinements for Collection 6, J. Appl. Meteorol. Climatol., 51, 1145-1163, doi:10.1175/JAMC-D-11-0203.1.

Clough, S. A., Iacono, M. J., and Moncet, J.-L.: Line-by-line calculation of atmospheric fluxes and cooling rates: Application to water vapor, J. Geophys. Res., 97, 15761-15785, 1992.

Clough, S. A., Shephard, M. W., Mlawer, E. J., Delamere, J. S., Iacono, M. J., Cady-Pereira, K., Boukabara, S., and Brown, P. D.: Atmospheric radiative transfer modeling: A summary of the AER codes, J. Quant. Spectrosc. Radiat. Transfer, 91, 233-244, 2005.

---

## Author Comment (AC3) · 4 Apr 2016

The authors would like to offer a response to comments received via personal communication regarding the sensitivity of the present approach to in-cloud water vapor absorption. An important observation has been made that the channels within the 1.88µm water vapor absorption band used here are sensitive to water vapor absorption within the cloud layer itself, which, as has been correctly pointed out, is not explicitly accounted for in the cloud retrieval look-up tables nor in the above-cloud atmospheric correction. As was suggested to us, we performed a sensitivity analysis using high spectral resolution (0.1nm spacing) forward radiative transfer (RT) simulations that couple line-by-line column transmittances from LBLRTM with DISORT such that the eMAS 1.83µm and 1.93µm spectral response functions can be used to compute band-averaged TOA reflectances. The RT calculations were performed for a 1km thick ice cloud located at 12km altitude (an altitude similar to the case studies in the paper), varying COT (0.1 to 40), CER (5 to 55µm), cosine of the sensor zenith angle $\mu$ (0.45 to 1.0), cosine of the solar zenith angle $\mu_0$ (0.15 to 1.0), and relative azimuth angle (0 to 180°). Two RT simulations were performed, one with water vapor above- and in-cloud, and one with water vapor above-cloud only; in both cases the surface was assumed to be non-reflecting. We found the TOA reflectance bias between the two RT runs to be a function primarily of cloud optical thickness (COT) as expected due to increasing in-cloud path lengths with COT. While the absolute reflectance difference increases with COT, the relative difference decreases, with a maximum relative error of roughly 7-8% at COT=1; the mean relative bias is at roughly 1-2% at COT=40. This is shown in the figure below, in which mean (solid lines) and ±1σ (dotted lines) reflectance errors are plotted as a function of COT for 1.83µm (blue) and 1.93µm (red); relative reflectance errors are defined here as the difference (R[in+above]-R[above])/R[above], which is comparable to the atmospheric correction bias such that negative values imply an under-correction for atmospheric absorption (i.e., darker atmospherically-corrected reflectance) and thus smaller retrieved COT and larger retrieved CER. However, note that these biases are nevertheless smaller than the 10% radiometric uncertainty assumed for the 1.83µm and 1.93µm channels (see p. 12, line 279).

[Figure]

From the below plots showing COT and CER retrieval uncertainty from individual uncertainty components (data are from the two SEAC[4]RS case studies shown in the manuscript), it is evident that a 10% error (i.e., the blue line denoting radiometric uncertainty) corresponds to COT uncertainty of roughly 10-20% over much of the observed COT distribution (though much larger at larger COT), and CER uncertainty of roughly 20-30%. While these uncertainties are large, it should be reemphasized that 10% uncertainty is larger than the TOA reflectance sensitivity to in-cloud water vapor absorption (as stated above, roughly 7-8% at its peak); in fact, at COT=1 the

10% radiometric uncertainty yields a COT retrieval uncertainty of about the same magnitude (CER retrieval uncertainty is approximately double). Thus explicitly ignoring in-cloud water vapor absorption in the present retrievals should be expected to result in smaller retrieval errors than those that result from the radiometric uncertainty (and in fact considerably smaller errors for much of the COT solution space). However, unlike radiometric uncertainty, we acknowledge that the in-cloud water vapor absorption error source is expected to involve a bias (i.e., smaller COT, larger CER) over much of the COT space in addition to a random component.

[Figure]

That said, in-cloud absorption is implicitly accounted for at least partially by the above-cloud atmospheric correction process, assuming the radiative cloud top height retrieved from the thermal IR channels is located below the physical cloud top observed by the lidar. Thus the path length from TOA to the radiative cloud top is expected to include part of the cloud layer itself. This is typically the case when using the heritage 11µm IR-window and 13µm $CO_2$-slicing cloud top retrieval techniques (see, e.g., Holz et al., 2008), whose results will be similar to the OE-IR cloud top retrievals shown in Figs. 5 and 7 that use identical spectral information. We note, however, that for the present investigation we use the cloud top retrievals from the NOAA AWG PATMOS-x algorithm, consistent with the archived eMAS cloud products that were produced for the SEAC[4]RS field campaign at the time of this writing. These retrievals are, at least for the case studies shown here, near the cloud top observed by CPL. Nevertheless, to the extent that the AWG PATMOS-x retrievals provide a radiative cloud height that is below the physical cloud top, their use can at least partially offset the impacts of explicitly ignoring the in-cloud water vapor absorption in the forward-calculated LUTs and atmospheric correction. Finally, we note that in practice it is impractical to estimate the exact in-cloud water vapor absorption (or the errors resulting from its neglect) at pixel-level due in part to the lack of a computationally efficient on-line RT algorithm that necessitates the use of pre-computed LUTs, as well as the general ignorance of the retrieval algorithm to pixel-level radiative cloud top retrieval biases. We have added to the manuscript a brief summary of the above discussion of the in-cloud water vapor absorption

sensitivity (p. 10-11, lines 235-259), as well as other details regarding the above-cloud atmospheric correction process (p. 10, lines 218-230).

**References**

Holz, R. E., Ackerman, S. A., Nagle, F. W., Frey, R., Dutcher, S., Kuehn, R. E., Vaughan, M. A., and Baum, B.: Global Moderate Resolution Imaging Spectroradiometer (MODIS) cloud detection and height evaluation using CALIOP, J. Geophys. Res., 113, doi:10.1029/2008JD009837, 2008.